# Body Adiposity Partially Mediates the Association between *FTO* rs9939609 and Lower Adiponectin Levels in Chilean Children

**DOI:** 10.3390/children10030426

**Published:** 2023-02-22

**Authors:** Carolina Ochoa-Rosales, Lorena Mardones, Marcelo Villagrán, Claudio Aguayo, Miquel Martorell, Carlos Celis-Morales, Natalia Ulloa

**Affiliations:** 1Latin American Brain Health Institute (BrainLat), Universidad Adolfo Ibáñez, Santiago 7941169, Chile; 2Centro de Vida Saludable, Universidad de Concepción, Concepción 4070374, Chile; 3Laboratorio de Ciencias Biomédicas, Facultad de Medicina, Universidad Católica de la Santísima Concepción, Concepción 4090541, Chile; 4Departamento de Bioquímica Clínica e Inmunología Facultad de Farmacia, Universidad de Concepción, Concepción 4070386, Chile; 5Departamento de Nutrición y Dietética, Facultad de Farmacia, Universidad de Concepción, Concepción 4070386, Chile; 6British Heart Foundation Glasgow Cardiovascular Research Centre, Institute of Cardiovascular and Medical Sciences, College of Medical, Veterinary and Life Sciences, University of Glasgow, Glasgow G12 8QQ, UK; 7Laboratorio de Rendimiento Humano, Grupo de Estudio en Educación, Actividad Física y Salud (GEEAFyS), Universidad Católica del Maule, Talca 3466706, Chile; 8Centre of Exercise Physiology Research (CIFE), Universidad Mayor, Santiago 7500994, Chile

**Keywords:** adiponectin, inflammation, obesity, *FTO* variant, children obesity, mediation analysis

## Abstract

Children carrying the minor allele ‘A’ at the fat mass and obesity-associated protein (*FTO)* gene have higher obesity prevalence. We examined the link between *FTO* rs9939609 polymorphism and plasma adiponectin and the mediating role of body adiposity, in a cross-sectional study comprising 323 children aged 6–11 years. Adiponectin and *FTO* genotypes were assessed using a commercial kit and a real-time polymerase chain reaction with high-resolution melting analysis, respectively. Body adiposity included body mass index z-score, body fat percentage and waist-to-hip ratio. To investigate adiponectin (outcome) associations with *FTO* and adiposity, linear regressions were implemented in additive models and across genotype categories, adjusting for sex, age and Tanner’s stage. Using mediation analysis, we determined the proportion of the association adiponectin-*FTO* mediated by body adiposity. Lower adiponectin concentrations were associated with one additional risk allele (β_additive_ = −0.075 log-μg/mL [−0.124; −0.025]), a homozygous risk genotype (β_AA/TT_ = −0.150 [−0.253; −0.048]) and a higher body mass index z-score (β = −0.130 [−0.176; −0.085]). Similar results were obtained for body fat percentage and waist-to-hip ratio. Body adiposity may mediate up to 29.8% of the *FTO*-adiponectin association. In conclusion, *FTO* rs9939609-related differences in body adiposity may partially explain lower adiponectin concentrations. Further studies need to disentangle the biological pathways independent from body adiposity.

## 1. Introduction

In the last decades, overweight and obesity prevalence has increased worldwide, affecting adults, children and adolescents in developed and developing countries [1]. In Chile, childhood obesity has steadily risen since the late 1980’s [2,3]. By 2020, prevalence of childhood overweight and obesity reached up to 64%, across different school grades [4]. At all ages the life course of obesity is associated with various detrimental health effects, such as metabolic syndrome (MetS) and type 2 diabetes (T2D). It is estimated that around 60% of those suffering from obesity in their early life will show at least one metabolic alteration related to non-communicable chronic diseases (NCDs) in their adulthood, such as hypertension, dyslipidemia, insulin resistance or metabolic syndrome [5,6,7].

Obesity, a condition of excessive body fat, has been linked to lower concentrations of plasma adiponectin, a type of adipokine secreted by the adipose tissue [8]. Adiponectin has received special attention due to its pleiotropic role and beneficial effects on tissues such as skeletal muscle, liver, heart, and kidney tissues [9]. Several studies have provided evidence of adiponectin’s anti-inflammatory, anti-atherogenic [10,11] and insulin-sensitizing effects [12,13,14]. Moreover, lower plasma adiponectin has been observed among obese and T2D individuals [15,16], which has drawn more attention as a promising therapeutic target against T2D and cardio-metabolic traits.

Although there are several behavioral risk factors that contribute to obesity, genetic load plays a relevant role in the etiology of obesity. Since 2007, more than 1100 independent loci from almost 60 genome-wide association studies (GWAS) have been identified in association with several obesity traits [17]. Among them, the strongest susceptibility gene identified is the Fat Mass and Obesity (*FTO*)-associated gene [18]. The relationship between obesity and the minor allele ‘A’ of the *FTO* single-nucleotide polymorphism (SNP) rs9939609 is well established [18,19] among adults and children from diverse ethnic backgrounds, including Chileans [20,21,22,23,24,25]. For example, a study in Caucasian adults found that carriers of the A-allele had higher odds of overweight and obesity by 19% (95% CI 1.06; 1.20) and 27% (1.20; 1.34), respectively [23]. Moreover, the *FTO* rs9939609 risk variant has been suggested to predispose carriers to obesity-related diseases, such as T2D, and this association may be explained by body adiposity [23,26,27].

The link between the *FTO* rs9939609 risk variant and plasma levels of adiponectin is less clear. One study reported that adults carrying the rs9939609 SNP risk variant had no differences in adiponectin circulating levels [28], while others showed that A allele carriers had significantly lower adiponectin concentration [21,29], which was attenuated after BMI adjustment, suggesting that such association might occur through changes in body adiposity [29].

The evidence from studies in populations of children is limited. Therefore, we sought to research the association of plasma adiponectin concentration with the *FTO* rs9939609 genotypes (AA, TA, and AA) and measurements of general and central adiposity in Chilean children. In addition, we investigated the potential role of adiposity markers as mediators in the association between *FTO* rs9939609 and adiponectin.

## 2. Materials and Methods

### 2.1. Study Design and Population

This cross-sectional study included a sample of children living in the Biobío Region of Chile. Participants were 6 to 11 years old and free of any chronic disease. Participants with incomplete data on anthropometric and adiposity measures, *FTO* genotype and circulating adiponectin levels were excluded from the study. Out of the 361 recruited individuals, 37 participants with missing data on body fat percentage (BF%) and one individual with adiponectin blood concentration beyond five standard deviations were removed from the analyses. The final analytical sample size was n = 323.

### 2.2. Plasma Determinations

A sample of 4 mL of fasting peripheral blood was collected. Plasma adiponectin was determined using a commercial ELISA kit (Linco Research, St. Charles, MO, USA) and a multi-reader (Synergy 2, Biotek, Winooski, VT, USA).

### 2.3. Anthropometric Measurements

Data on weight (kg), height (cm), body mass index (BMI, kg/m^2^), waist–height index (WHtR) and BF% were collected using an anthropometrics manual [30]. Briefly, weight was assessed on light clothing and without shoes on a Tanita scale (TANITA TBF-300, TANITA, Tokyo, Japan; 1 g accuracy). Height was measured using wall-mounted stadiometers (Seca, model 208, 0.1 cm precision). The waist circumference was measured at the midpoint between the last rib and the upper border of the iliac crest with a non-elastic flexible tape (Seca, model 201, accuracy 0.1 cm). Body composition (body lean mass, body fat mass and BF%) was determined using a bioelectrical impedance analysis (TANITA TBF-300, Tokyo, Japan). The pubertal stage was established by a pediatrician according to the Tanner criteria [31]. BMI z-score was computed following the WHO definitions. Body weight was divided by height in meters squared and then normalized based on age and sex. Nutritional status was classified, as normal weight (z-score BMI > −2SD and <+1SD); overweight (z-score BMI > +1SD or <+2SD); obesity (z-score BMI > +2SD), as proposed by WHO [32].

### 2.4. Identification of Allelic Variants of FTO rs9939609 Polymorphism

We used the Mini Kit QIAamp DNA Blood (Qiagen GmbH, Hilden, Germany) to extract genomic DNA from leukocytes following the manufacturer’s instructions. The DNA amplification by real-time polymerase chain reaction (PCR) amplifications and the high-resolution melting analysis (HRM) were performed with the thermocycler Rotor-Gene 6500 (Corbett Research, Sydney, Australia). Briefly, 100 ng/µL of genomic DNA were incubated at 95 degrees Celsius (C) for 10 min with 3.0 mM magnesium, 12.5 µL SensiMix HRM (Quantace) reagent, 1 µL EvaGreen dye and 600 nM primers. It was followed by 40 cycles at 95 C for 15 s, 59 C for 10 s and 72 C for 10 s following a standardized protocol [33]. We based our primer selection on the work by López-Bermejo and colleagues. We used forward primer: 5′-AACTG GCTCTTGAATGAAATAGGATTCAGA-3′ and reverse primer: 5′-GTGATGCACTTGGATAGTCTCTGTTACTCT-3′ [34]. A graphic representation of PCR product sequence of 182 base pairs can be found in Appendix A. Wild type (TT), heterozygous (TA) and homozygous for the mutation (AA) genotypes were identified using the HRM method, which allows variations in single nucleic acids to be identified by detecting small differences in the DNA melting temperature [35]. Our HRM analysis investigated melting temperatures from 70 to 85 C, with increments of 0.1 C in each PCR cycle [33]. PCR melting curves were obtained using the Rotor-Gene 6500-incorporated software and compared with the melting curves from samples with known sequence (controls), using 95% confidence interval. The melting curves of TT, TA and AA *FTO* genotypes are shown in Appendix A. During HRM technique standardization, 20 random samples underwent sequencing (Faculty of Biological Sciences, Pontificia Universidad Católica de Chile) in order to confirm genotypes and be used as controls. As an example, Appendix A displays the sequencing data of one control sample. We carried out a 3% agarose gel electrophoresis in order to confirm the presence of a single PCR product. All samples were analysed in duplicates, with a 98% genotyping success rate.

### 2.5. Ethics

We adhered to the Declaration of Helsinki (1964), the Convention of the Council of Europe regarding human rights and biomedicine (1997), and the Universal Declaration on the human genome and human rights (UNESCO, 1997). Moreover, we met the requirements of the Chilean legislation in the field of biomedical research, data privacy and bioethics, according to Decree No. 114 of 2010, Law No. 20,120, and Decree update on 14 January 2013. In addition, this study protocol was approved by the Bioethics Committee of the Vice-Rectory of Research of Universidad de Concepción with number 352-2019 and date January 2019.

### 2.6. Statistical Analyses

The study population characteristics were presented as mean and standard deviation (SD), or as median and interquartile range (IQR) for continuous variables with normal or non-normal distribution, respectively. Absolute and relative frequency was used to describe categorical variables.

Linear regressions were used to investigate associations between circulating concentrations of adiponectin (dependent variable) and various measurements of general and central adiposity: BMI z-score, BF % and waist-to-height ratio (WHtR), as exposures (independent variables). To account for potential confounders, three statistical models were used. Model 1 was unadjusted, Model 2 was adjusted for sex, age and Tanner’s stage, and Model 3 added an adjustment for the *FTO* SNP rs9939609 genotype. The *FTO* SNP rs9939609 genotype was coded according to an additive model where the *FTO* genotype was coded as 0 = TT (homozygous for the wild type allele); 1 = TA (heterozygous for risk allele); 2 = AA (homozygous for the risk allele). The Chi-square test was used to estimate the Hardy–Weinberg equilibrium of the *FTO* alleles. Blood adiponectin concentrations were transformed to their natural logarithm to approximate normal distribution. The results were expressed in beta estimates and 95% lower and upper confidence intervals (β, 95% CI), for the variation in log-transformed adiponectin concentrations in µg/mL, per one unit increase in adiposity measures.

Next, we studied the relationship between adiponectin concentrations and the variants of the *FTO* SNP rs9939609 genotype in linear associations using statistical Models 1 and 2. For the independent variable, we followed two approaches: (1) as a numerical variable according to the additive model (0 = TT; 1 = AT; and 2 = AA); and (2) as genotype categories, comparing carriers of the AT or AA genotype with the TT genotype, respectively, thus using the wild type as the reference group. Further, a third model was performed with additional adjustment for a marker of general adiposity, BMI z-score. The results were expressed in beta estimates and 95% lower and upper confidence intervals (β, 95% CI), for the variation in log-transformed adiponectin, per one additional risk allele (approach 1), or among carriers of AT or AA, compared with TT carriers, respectively (approach 2).

Finally, we used model 3 to interrogate the role of body adiposity as a potential mediator in the association between adiponectin concentrations and the *FTO* SNP rs9939609 genotype, using mediation analysis from the mediation package in R [36]. This analysis dissects the total effect of the exposure (presence of one additional *FTO* rs9939609 risk allele) on the outcome (adiponectin concentration) into the direct and indirect effects. The direct effect is the part of the effect that goes directly or through mediators other than those currently studied. The indirect effect represents the portion of the effect that goes via (is mediated by) the variable under study (adiposity markers); then, the proportion mediated is quantified and expressed in percentage mediated. Further, this analysis works under the sequential ignorability assumption, which assumes no unmeasured confounding. Note that, despite that the notation in the mediation analysis uses the word ‘effect’, their results must not be interpreted as causal, given the observational and cross-sectional design of this study. Quasi-Bayesian confidence intervals were constructed for the estimated effects with 5000 simulations. Results are expressed as proportion mediated in percentage and 95% Cis. A schematic representation of the mediation analysis concept is displayed in Figure 1. All analyses were performed using R statistical software v.4.0.1 (R Foundation, Vienna, Austria).

## 3. Results

### 3.1. Sample Description

The study population’s characteristics are shown in Table 1. Briefly, participants were on average 8.8 (SD ± 2.2) years old, were more often at pre-pubertal stage (77.4%) and female (50.8%), and they lived in the Biobío Region, Chile.

### 3.2. Associations between Circulating Adiponectin and FTO rs9939609

Table 2 shows the inverse association between the presence of one additional risk allele at SNP rs9939609 and adiponectin concentration (β_additive_= −0.075 [−0.124; −0.025], *p* = 0.003) controlling for age, sex and Tanner’s stage. For the *FTO* genotype, only the homozygous for the risk allele (AA) showed an association with adiponectin concentrations (β_AA vs. TT_ = −0.150 [−0.253; −0.048], *p* = 0.004). This association remained after adjusting the analysis by BMI (β = −0.104 [−0.101; −0.004], *p* = 0.041) (Table 2 and Figure 2).

### 3.3. Link between Circulating Adiponectin and Body Adiposity

Using Model 2, we found that natural log-transformed adiponectin levels in blood were inversely associated with various measures of overall (BMI z-score and total BF %) and central obesity (WHtR), independent of sex, age and Tanner’s stage. A one-unit increase in BMI z-score and one-percent increase in total body fat were associated with lower adiponectin concentration (β = −0.130 log μg/mL [95% CI −0.176; −0.085], *p* = 4.63 × 10^−8^ and β= −0.012 [−0.016; −0.007], *p* = 2.38 × 10^−7^, respectively) (Table 2 and Figure 3). Similarly, the increase in one unit of WHtR was related to lower adiponectin (β = −1.386 [−1.959; −0.814], *p* = 2.91 × 10^−6^) (Table 2). After adjusting for the *FTO* SNP rs9939609, the associations remained (Table 2).

### 3.4. Body Adiposity as Partial Mediator of the Adiponectin–FTO rs9939609 Association

Further, we investigated the potential role of general and central adiposity markers as mediators in the *FTO* genotype–adiponectin associations. In the mediation analysis, we observed that higher adiposity measures of BMI, WHtR and BF%, respectively, mediate a proportion from 23.9% (6.8; 68.0, P_mediation_ = 0.006, for WHtR) to 29.8% (10.4; 79.0, P_mediation_ = 0.003, for Z-score BMI) of the total effect of one additional *FTO* rs9939609 risk allele on plasma adiponectin levels, called the indirect effect (Table 3).

## 4. Discussion

In this population of Chilean children, we found associations between lower plasma adiponectin and the presence of one additional A allele at the *FTO* rs9939609 polymorphism, and across *FTO* genotypes. Moreover, inverse associations between adiponectin and measures of general and central adiposity markers (BMI Z-score, WHtR and BF%) were observed. Furthermore, we quantified the proportion of the adiponectin–*FTO* association explained by body adiposity differences, finding that up to 30% of the association is mediated by *FTO*-related levels of BMI Z-score, WHtR or BF%.

Adiponectin is an endocrine factor secreted by adipocytes [37] and displays beneficial effects on various tissues, such those related to glucose uptake and fatty acid metabolism [38,39]. Additionally, lower adiponectin concentrations are related to intermediate risk factors for T2D, such as higher blood glucose, insulin, and triglycerides [33,40].

Experimental studies show that plasmatic adiponectin and other hormones regulating food intake and satiety regulation, such as leptin and ghrelin, are associated with *FTO* polymorphisms [40]. Several observational studies in populations of diverse ethnic backgrounds (Caucasian, Mexican, Turkish, Indian and Chinese) have investigated the relationship between *FTO* rs9939609 and plasma adiponectin, reporting significant associations [20,21,22,23,24,25]. In agreement, our study in a pediatric Chilean population found a significant relationship between lower concentrations of adiponectin and the presence of one additional A allele at rs9939609, in multivariate regressions adjusted for age, sex and Tanner’s stage. Similar results were found among AA carriers, as compared to wild-type (TT) carriers, while no significant results were found when comparing TA with TT carriers. Nevertheless, our results contradict studies carried out in Tunisian [41] and Iranian [28] adults and in Romanian children [42], which failed to find significant differences in adiponectin levels across *FTO* rs9939609 genotypes. Moreover, a randomized controlled trial on a two-year calorie restriction revealed that the presence of the A allele did not influence adiponectin levels in response to the intervention, although they found significant associations at baseline [43] similar to other studies [44,45].

Mechanisms underlying the potential relationship *FTO*-adiponectin have been elucidated to a limited extent. On the one hand, lower plasma adiponectin concentration has been observed among obese or higher BMI adults [46,47] and among obese children [48,49], in line with our results. On the other hand, adiposity excess and metabolic syndrome is well known to be prevalent among carriers of the A allele at rs9939609 [18,19]. Moreover, we previously reported significant associations of rs9939609 with higher body adiposity [20], obesity [33] and MetS prevalence in Chilean children, finding the strongest effect among AA carriers [50].

The current study provides additional evidence on the role of body adiposity in the observed associations using two approaches. Firstly, additional adjustment of the association adiponectin-*FTO* for BMI z-score resulted in an attenuated beta estimate size, although the associations remained significant. This observation is in line with previous studies [21,22,23,24,25,26,27,28,29] which performed multivariate analysis and found attenuation of the association after accounting for body adiposity measures. Further, we implemented a more advanced method, mediation analyses, in order to compute *FTO*’s indirect effect on adiponectin levels, defined as the effect that goes through the mediator (body adiposity). We found that, of the total effect of *FTO* on adiponectin concentrations, from 23.9% (WHtR) to 29.8% (BMI z-score) is mediated by *FTO*-related differences in general or central body adiposity, with the largest effect attributed to the BMI z-score. Nevertheless, there is still a major part of the association that is not explained by *FTO*-related adiposity, but by other biological pathways beyond the scope of this study. Given the possible role of adiponectin as a promising therapeutic target against highly prevalent metabolic diseases such as T2D [51], future studies aiming to unravel mechanisms related to *FTO* rs9939609 regulating plasma adiponectin concentrations different from body adiposity are warranted.

Based on the available evidence, part of the unexplained effect might be determined by the gene-environment interactions. Some studies suggest that *FTO* polymorphism’s effect may be modulated by lifestyle factors [52], such as diet quality and physical exercise [21,53]. For example, macronutrient–gene interactions might affect obesity phenotypes, potentially by regulation of *FTO* and *IRX3* gene expression [54]. Recent research shows that adherence to a Mediterranean dietary pattern, higher intake of unsaturated fatty acids, whole grains, polyphenols and probiotics improves the inflammatory biomarker profile, including the elevation of plasma adiponectin [55,56]. Moreover, it is well known that physical exercise triggers a number of signaling pathways stimulating the production of the bioactive molecules, adiponectin among them, that exert beneficial health effects [57]. A study reported that among individuals with the lowest physical activity levels, those carrying the A allele at rs9939609 had lower plasma adiponectin concentrations as compared to carriers of the TT genotype [21]. More studies are needed to dissect the extent to which environmental factors may contribute to attenuating the genetic predisposition to adverse metabolic health outcomes conferred by *FTO* risk genotypes.

Our study has some strengths. We used a well-characterized children population, including details on their pubertal stage as a biochemical marker of obesity. In addition, we used a formal mediation analysis test, an advanced statistical method to compute the proportion of the effect. Among the weaknesses of our study, we count a study population of a restricted size, although we were able to detect significant associations. Further, we used classic anthropometric and bio-impedance measures of body fat, while more accurate methods for body composition are preferable, such as dual-energy X-ray absorptiometry and computed tomography [58]. Moreover, we had limited data in lifestyle factors and other genetic polymorphisms regulating adiponectin production, such as ADIPOQ or genetic ancestry. About the latter, a previous work from our research group performed the ethnic characterization of the same study sample based on Amerindian haplogroups assessed in mitochondrial DNA [33]. The authors observed that 85% had Amerindian lineages, and that among normal-weight and obese children, the proportion of non-Amerindian individuals did not differ per group. Thus, we do not expect that genetic ancestry affects our results.

Further, the cross-sectional design of our study restrains the mediation analysis interpretation, given that the studied mediators were collected at the same time point as blood collection for adiponectin measurement. In addition, it contains some strong assumptions, such as the ignorability assumption, wherein no potential unmeasured confounding is present [36]. Although we addressed the relevant confounders, unmeasured confounding should not be ruled out. For this reason, the results from our observational study must not be interpreted as causal; however, they do provide evidence to inspire future well-powered studies and the use of causal inference methods. Finally, these results are not generalizable, and replication of the findings to other populations merits further research.

## 5. Conclusions

To the extent of our knowledge, this is the first study to formally suggest that the association between adiponectin and the *FTO* risk variant might be partly mediated by changes in body adiposity induced by *FTO* rs9939609 in a Chilean children population. Further research is needed to unravel the biological pathways linking SNP rs9939609 and adiponectin independently from body adiposity. More studies using accurate methods to assess body fat and a causal design are needed confirm these findings. This study enlarges the body of evidence and confirms previous findings embedded in populations with diverse ethnic backgrounds and age groups.

## Figures and Tables

**Figure 1 children-10-00426-f001:**
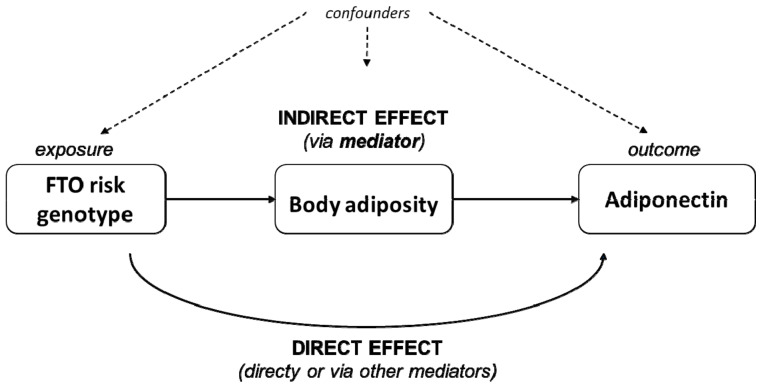
Schematic representation of the causal mediation analysis. The total effect of an exposure (*FTO* genotype) over an outcome (concentration of circulating adiponectin) is dissected into direct and indirect effects. The indirect effect is the one being exerted via a potential mediator of interest (body adiposity). The remaining part of the effect goes directly or via mediators other than the ones under current study.

**Figure 2 children-10-00426-f002:**
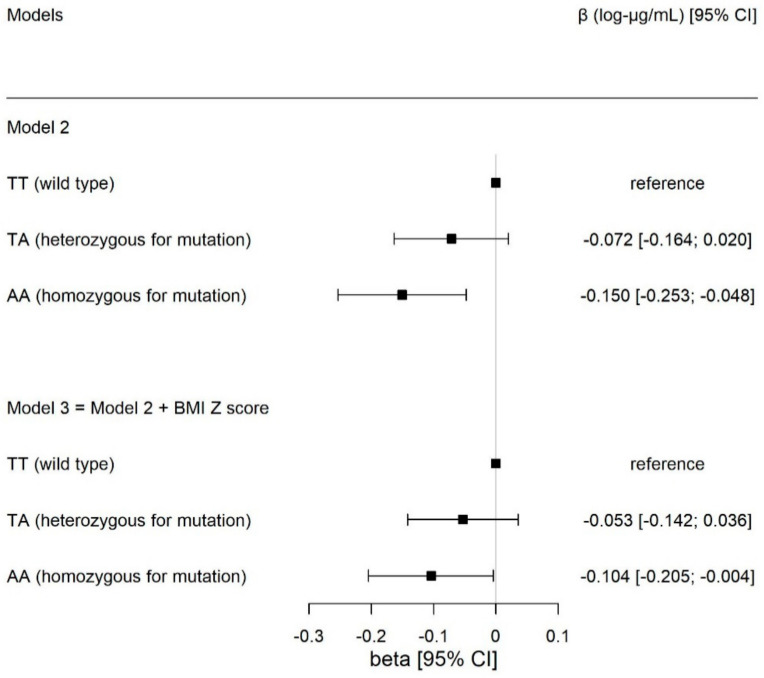
Associations between circulating adiponectin concentrations in log−transformed μg/mL, with *FTO* genotype for SNP rs9939609, comparing carriers of the heterozygous (TA) and homozygous (AA) genotype for the mutation, with the wild-type individuals (TT, reference group), respectively. The results are expressed in beta effect estimates of and their respective 95% CIs. Model 2 was adjusted for age, sex, and Tanner’s stage. Model 3 included the adjustments in model 2 plus BMI z-score.

**Figure 3 children-10-00426-f003:**
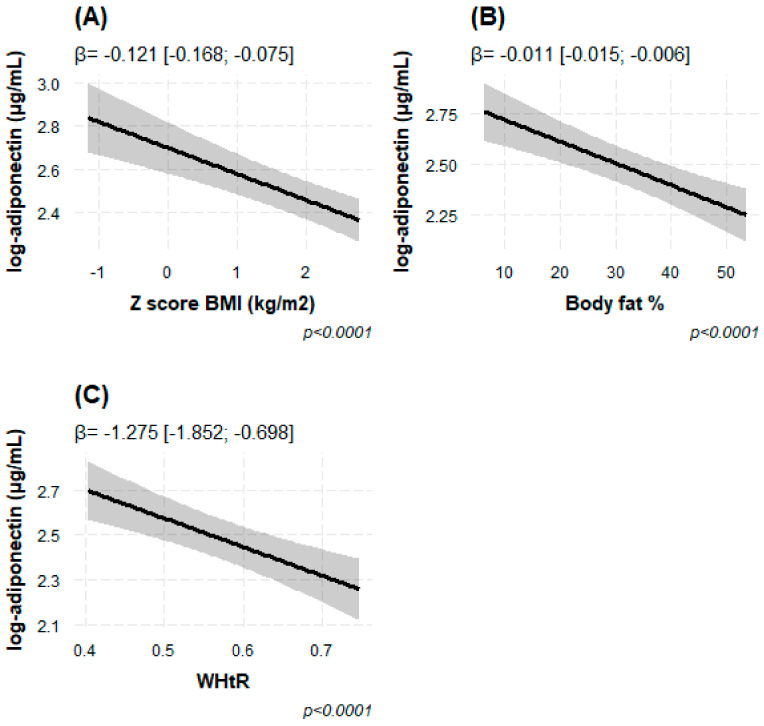
(**A**–**C**) Associations of circulating adiponectin concentrations in log−transformed μg/mL, with several measures of overall and central adiposity. (**A**) BMI z-score, (**B**) total body fat % and (**C**) Waist to height ratio (WHtR). The statistical model was adjusted for age, sex, Tanner’s stage and *FTO* rs9939609 genotype. The results are expressed in beta effect estimates of and their respective 95% CIs.

**Table 1 children-10-00426-t001:** Characteristics of the study population.

Variable	Overall Study Population	TT	TA	AA
Total (n)	323	167	91	65
Age, years, mean (SD)	8.83 (1.28)	8.82 (1.29)	8.59 (1.22)	8.91 (1.32)
Females, n (%)	164 (50.8)	88 (52.7)	43 (47.3)	33 (50.8)
Tanner’s stage				
Pre-pubertal, n (%)	250 (77.4)	127 (76.0)	75 (82.4)	48 (73.8)
Pubertal, n (%)	73 (22.6)	40 (24.0)	16 (17.6)	17 (26.2)
Height, cm, median [IQR]	133 [126, 140]	133 [126, 140]	132 [127, 139]	137 [128, 143]
Weight, kg, median [IQR]	37.9 [30.9, 47.4]	37.2 [29.8, 45.5]	38.5 [32.5, 44.7]	43.8 [34.1, 52.8]
Waist circumference, cm, median [IQR]	73.5 [64.6, 81.0]	70.8 [63.8, 80.3]	74.0 [66.0, 80.6]	77.0 [70.0, 83.3]
Body mass index, kg/m^2^ median [IQR]	22.2 [17.9, 24.5]	21.3 [17.6, 23.9]	22.3 [17.7, 24.2]	23.8 [20.5, 26.1]
z-score BMI, median [IQR]	1.83 [0.69, 2.12]	1.78 [0.62, 1.99]	1.86 [0.64, 2.17]	2.01 [1.75, 2.23]
Waist circumference to height ratio, median [IQR]	0.56 [0.49, 0.60]	0.55 [0.48, 0.59]	0.57 [0.49, 0.60]	0.57 [0.53, 0.61]
Body fat mass, %, mean [IQR]	30.2 [21.6, 36.2]	28.4 [21.3, 34.9]	29.5 [21.3, 36.1]	34.8 [24.8, 37.8]
Adiponectin, µg/mL, median [IQR]	14.2 [10.58, 17.12]	14.2 [10.9, 18.4]	13.0 [10.6, 16.8]	12.4 [9.7, 15.3]

**Table 2 children-10-00426-t002:** Associations between adiponectin (log-transformed, µg/mL) and several markers of general and central adiposity and the *FTO* rs9939609 genotype.

		Model 1			Model 2		Model 3 ^a,b^		
Independent Variables	β	95% CI	*p*	β	95% CI	*p*	Β	95% CI	*p*
FTO genotypes ^a^									
TT (wild type)	Ref.			Ref.			Ref.		
TA	−0.069	(−0.161; 0.022)	0.137	−0.072	(−0.164; 0.020)	0.127	−0.053	(−0.142; 0.036)	0.243
AA (mutant)	−0.150	(−0.253; −0.048)	**0.004**	−0.150	(−0.253; −0.048)	**0.004**	−0.104	(−0.205; −0.004)	**0.041**
Additive	−0.074	(−0.124; −0.025)	**0.003**	−0.075	(−0.124; −0.025)	**0.003**	−0.052	(−0.101; −0.004)	**0.035**
Adiposity phenotypes ^b,c^									
BMI Z score	−0.130	(−0.176; −0.085)	**4.63 × 10^−8^**	−0.130	(−0.176; −0.084)	**5.10 × 10^−8^**	−0.121	(−1.677; −0.075)	**4.50 × 10^−7^**
WHtR	−1.393	(−1.957; −0.828)	**1.90 × 10^−6^**	−1.386	(−1.959; −0.814)	**2.91 × 10^−6^**	−1.275	(−1.852; −0.698)	**1.85 × 10^−5^**
BF%	−0.011	(−0.015; −0.007)	**2.66 × 10^−7^**	−0.012	(−0.016; −0.007)	**2.38 × 10^−7^**	−0.011	(−0.015; −0.006)	**2.00 × 10^−6^**

Model 1: unadjusted model; Model 2 adjusted for: age, sex and Tanner’s stage. ^a^ Model 3: model 2 additionally adjusted for BMI z score when the independent variables were *FTO* genotypes. ^b^ Model 3: model 2 additionally adjusted for *FTO* genotype when the independent variables were adiposity phenotypes. ^c^ For the analyses on the several adiposity phenotypes, a Bonferroni corrected threshold of 0.05/3 = 0.017 was set to test statistical significance. Bold text indicates statistical significance of *p* < 0.05 when the independent variables were FTO genotypes, or *p* < 0.05 when the independent variables were adiposity phenotypes.

**Table 3 children-10-00426-t003:** Mediation analysis for the association between adiponectin and *FTO* rs9939609 genotype, with several measures of adiposity modelled as potential mediators.

Potential	Average Mediation Effect			Total Effect			Proportion Mediated		
Mediators	β	95% CI	*p*	β	95% CI	*p*	%	95% CI	*p*
BMI Z-score	−0.022	(−0.041; −0.010)	**0.001**	−0.074	(−0.125; −0.030)	**0.002**	29.8	(10.4; 79.0)	**0.003**
WHtR	−0.018	(−0.035; 0.000)	**0.003**	−0.074	(−0.125; −0.030)	**0.003**	23.9	(6.8; 68.0)	**0.006**
BF%	−0.022	(−0.040; −0.010)	**0.003**	−0.074	(−0.125; −0.030)	**0.002**	28.6	(9.3; 77.0)	**0.005**

Mediation analysis was performed using Model 3, adjusted for age, sex and Tanner’s stage. Bonferroni corrected threshold of 0.05/3 = 0.017 was set to test statistical significance. Bold text indicates statistical significance at *p* < 0.017.

## Data Availability

The data presented in this study are available on request from the corresponding author.

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
