# Peer review of "Body Adiposity Partially Mediates the Association between *FTO* rs9939609 and Lower Adiponectin Levels in Chilean Children"

_children, 2023, doi:10.3390/children10030426_

Round 1
Reviewer 1 Report
-acronym in the abstract should be avoided
- the abstract is not informative, it is not clear wether authors recruited children, how many, the methods etc.
- the great limitation of the study is that the authors cannot exclude that adiponectin modulation is due to polymorphisms in ADIPOQ gene
- inclusion and exclusion criteria should be specified
Reviewer 2 Report
The entire manuscript would benefit significantly from a grammatical revision.
There is an error in participants' age range, i.e, in the abstract it stated, children aged 6–11 years but in the "Study Design and Population" section it stated as 7 to 11 years old.
It left me bewildered.
The authors should consider the potential confounders in their statistical analysis, such as genetic ancestry.
It is necessary that the authors describe the detail of band length for SNP with added a PCR product figure.
A part of the samples must be sequenced and shown in figure.
Round 2
Reviewer 1 Report
the authors have improved the manuscript appropriately
Reviewer 2 Report
Accept in current form